# Impact of Nano–Sized Polyethylene Terephthalate on Microalgal–Bacterial Granular Sludge in Non–Aerated Wastewater Treatment

Chao Du, Wenxuan Xiong, Guangya Zhu and Bin Ji *

Department of Water and Wastewater Engineering, School of Urban Construction, Wuhan University of Science and Technology, Wuhan 430065, China; duchao22@163.com (C.D.); 15872055353@163.com (W.X.); guangya_zhu@163.com (G.Z.)
* Correspondence: binji@wust.edu.cn; Tel.: +86-27-68893616

**Abstract:** The widely used plastics in our daily lives have resulted in ubiquitous microplastics and nanoplastics in wastewater, such as polyethylene terephthalate (PET). As an emerging green process for wastewater treatment and resource recovery, microalgal–bacterial granular sludge (MBGS) aligns with the concept of the circular economy. However, it is unclear whether the tiny PET can affect the MBGS process. Thus, this study investigated the impact of nano–sized PET (nPET) on the MBGS process. The results showed that 10 to 30 mg/L nPET had no obvious impact on pollutant removal as compared with the control group. However, the performance of the MBGS with the addition of 50 mg/L nPET became worse after 15 days. Scanning electron microscopy revealed that the MBGS adsorbed nPET by generating extracellular polymeric substances. Further microbial analyses showed that the algal abundance in prokaryotes slowly declined with increasing concentrations of nPET, while the reduced energy storage and electron transfer in eukaryotes might lead to an inferior performance at 50 mg/L nPET. Overall, the MBGS was demonstrated to exhibit good adaptability to nPET–containing wastewater, which showed the potential to be applied for the treatment of municipal wastewater containing nanoplastics.

**Keywords:** municipal wastewater; microalgal–bacterial granular sludge; polyethylene terephthalate; circular economy; resource recovery; nanoplastics

## 1. Introduction

Currently, to cope with the energy crisis and climate change, novel wastewater treatment processes are urgently needed to improve the efficiency of wastewater treatment and reduce operating costs while following the concept of the circular economy [1]. As an emerging process, the microalgal–bacterial granular sludge (MBGS) process has attracted great attention for its high efficiency, low energy consumption, minimal carbon dioxide emissions, and huge resource recovery potential [2,3]. In this process, microalgae can absorb carbon dioxide and nutrients, and generate oxygen through photosynthesis, and the resulting oxygen is then used by bacteria for organic oxidation in the wastewater to produce carbon dioxide [4]. Due to the above symbiotic effect, the non–aerated MBGS can effectively remove organics, nitrogen, and phosphorus, making it an energy–saving and efficient wastewater treatment process.

Emerging pollutants have caused unprecedented disasters in wastewater treatment plants (WWTPs) around the world, including microplastics (MPs), nanoplastics (NPs), etc. [5,6]. Plastic products have been widely used in all walks of life due to their excellent characteristics and low cost [7]. Due to a lack of awareness about environmental protection, discarded plastic products are decomposed into MPs smaller than 5 mm and NPs smaller than 1 μm which can be dissolved in water under natural weathering, biodegradation, photodegradation, and mechanical degradation, etc. [8,9]. As a result, MPs and NPs are

prevalent in natural water bodies like rivers, lakes, and oceans and in WWTPs through the social cycle of water [10]. Researchers have found that the concentration of NPs in ocean is about 0.51 mg/L, and these data are slowly increasing over time [11]. In recent years, studies on the toxicity of MPs have mainly focused on the synergistic effects with other pollutants, such as organic pollutants, etc. [12]. Studies have found that when MPs are exposed in a mixed form with pharmaceuticals and personal care products, they may be toxic to organisms [13]. It has also been reported that MPs increase the accumulation of cadmium in the livers, intestines, and gills of zebrafish, leading to oxidative damage and inflammation [14,15]. These phenomena indicate that the concentration of MPs has a negative effect on the growth and survival of organisms.

Polyethylene terephthalate (PET) is widely used in packaging materials for its excellent abrasive resistance, dimensional stability, and insulation [16]. The use of PET has made our lives more convenient, but improper handling of PET has caused serious damage to the environment. Despite the existence of PET recycling processes, a considerable amount of PET inevitably enters the WWTPs [17]. As a non–volatile solid, PET may adversely affect the rheological properties of sludge in WWTPs, and new industrial discharge processes in WWTPs have been proposed [18,19]. Previous studies have reported that PET powder, at a concentration of 200 mg/L, has a significant inhibitory effect on *Scenedesmus* sp., leading to increases in the concentrations of extracellular hydrogen peroxide and extracellular polymeric substances (EPS) in microalgae, and scanning electron microscope (SEM) images have shown that microalgae adhere to the surface of MPs and form heterogeneous aggregations [20]. However, it was reported that PET, at concentrations of 10–100 mg/L, has little effect on the growth of *Scenedesmus vacuolatus* [21]. Meanwhile, it was observed that the bacterial diversity decreased by 26.7% when PET was added to agricultural soil [22]. It is evident that PET may impact both algae and bacteria and that PET had a stronger effect on bacteria. However, the effect of PET on microalgal–bacterial symbiosis, such as the emerging MBGS, is still unclear.

To comprehend the impact of nano–sized polyethylene terephthalate on MBGS in non–aerated wastewater treatment, the present study aimed to (i) investigate the effects of different concentrations of nano–sized PET (nPET) on the removal of nitrogen, phosphorus, and the chemical oxygen demand (COD) in wastewater by the MBGS; (ii) analyze the toxicity of PET by measuring the EPS and chlorophyll (Chl) concentrations in the MBGS; and (iii) observe the interaction between nPET and MBGS through SEM and microbial communities. This study is expected to add knowledge on the feasibility of the MBGS process in the treatment and resource recovery of municipal wastewater containing NPs.

## 2. Materials and Methods

### 2.1. Compositions of Synthetic Wastewater

The synthetic wastewater mainly contained the following: 286 mg/L NaAc, 95.6 mg/L $NH_4Cl$, 13.2 mg/L $KH_2PO_4$, 10 mg/L $FeSO_4 \cdot 7H_2O$, 20 mg/L $CaCl_2$, 50 mg/L $MgSO_4 \cdot 7H_2O$, and 1.0 mL/L trace element solution [23]. The concentrations of COD, $NH_4^+$–N, and $PO_4^{3-}$–P were $400 \pm 10$, $25 \pm 1$, and $3 \pm 0.3$ mg/L, respectively, and the value of pH was approximately $7.3 \pm 0.3$.

### 2.2. Experimental Setup

Glass reactors (sealed cylindrical glass reactor which is 8.7 cm in height, 3.5 cm in diameter, 1.6 cm in neck diameter, 63 mL in total volume, and 50 mL in effective volume) were placed approximately 5 cm apart under an LED light (MBTL–T8–18, Hangzhou Mobate Biotechnology Co., Ltd., Hangzhou, China) with a light intensity of about 160 μmol/m$^2$/s on the surface of the reactors (12 h light/12 h dark). The initial volatile suspended solids (VSS) concentration of MBGS was about 5 g/L, and the 5 min sludge volume index (SVI$_5$) was about 41 mL/g. The nPET–containing wastewater was prepared using 200 mL volumetric flasks with varying concentrations of 0, 10, 20, 30, 40, and 50 mg/L nPET. The indoor temperature during the operation of the reactors was 20 °C, and the entire experiment

period was 15 days. The granule size of MBGS ranged from 0.80 to 1.25 mm. The reaction time for each cycle was set as 12 h. After a 12 h light cycle, the water was gathered from each reactor and filtered through 0.45 μm filters for further analysis.

### 2.3. Analytical Methods

COD, $NH_4^+$–N, $PO_4^{3-}$–P, $NO_3^-$–N, $NO_2^-$–N, VSS, and $SVI_5$ were measured according to the standard method [24], and ImageJ (version 1.41o, Java 1.6.0_10) was used to analyze the granule size of MBGS. The turbidity was measured by a portable turbidity meter (WGZ–1B, Hangzhou Qiwei Instrument Co., Ltd., Hangzhou, China). The value of pH and the dissolved oxygen (DO) concentration were determined by a pH meter (Ohaus, Parsippany, NJ, USA) and a DO meter (Yellow Springs, OH, USA), respectively. Optical microscopy (RX50, SOPTOP, Ningbo, China) and SEM (ThermoFisher, Waltham, MA, USA, Apreo S Hivac) were employed to observe the morphology of MBGS. The content of Chl was determined by acetone extraction [25]. EPS were extracted from MGBS by thermal extraction, and the contents of protein (PN) and polysaccharide (PS) were determined with the improved rapid Lowry method protein content determination kit (PRL002000, Shanghai Labaide Biotechnology Co., Ltd., Shanghai, China) and sulfuric acid anthrone colorimetry [23]. The extracted EPS was dried with a vacuum freeze dryer (FD–2) to obtain powder, and the surface functional groups and crystal structure of the EPS were analyzed by a Fourier transform infrared spectrometer (FTIR, Bruker, Billerica, MA, USA, INVENIO R), X–ray diffractometer (XRD, Rigaku, Tokyo, Japan, SmartLab SE), and X–ray photoelectron spectrometer (XPS, Shimadzu/Kratos, Manchester, UK, AXIS SUPRA+) [26]. Microbial community analysis was performed for the initial and final samples of MBGS based on Illumina Miseq sequencing [4], while the functional predictions of microorganisms were based on the Kyoto Encyclopedia of Genes and Genomes (KEGG) database. The atomic contents of the C and N elements were obtained by the full spectrum analysis using Avantage software (version 5.52). The experimental data were analyzed for variance using SPSS software (IBM SPSS Statistics 27), and the difference analysis results were significant at $p < 0.05$.

## 3. Results and Discussion

### 3.1. Pollutant Removal from nPET–Containing Wastewater

Based on Figure 1, the average removal efficiencies of COD, $NH_4^+$–N, and $PO_4^{3-}$–P were higher than 75%, 94%, and 81%, respectively, which indicates that MBGS performed well in treating nPET–containing wastewater. Compared with the control group without nPET, the removal of COD, $NH_4^+$–N, and $PO_4^{3-}$–P from the low concentration groups ($\leq$30 mg/L) was almost unaffected ($p > 0.05$). However, for wastewater containing nPET at a high concentration (50 mg/L), the removal of COD and $NH_4^+$–N exhibited varying degrees of decline, while COD and $NH_4^+$–N decreased by 6.10% ($p < 0.05$) and 2.57% ($p < 0.01$), respectively (Figure 1a,b). As for $PO_4^{3-}$-P (Figure 1c), nPET had little influence within the selected concentration range ($p > 0.05$), but there was a significant difference between 10 and 50 mg/L nPET ($p < 0.05$). $NO_3^-$–N and $NO_2^-$–N were undetectable, and $NH_4^+$–N was removed through microbial assimilation [27]. Since the performance of MBGS in the treatment of wastewater is closely related to photosynthesis [23], the alteration of water turbidity caused by the addition of nPET could affect the microalgae photosynthesis of MBGS [28,29]. As shown in Table 1, the turbidity of nPET–containing wastewater was linearly correlated with the concentration of nPET in the wastewater. For every 10 mg/L of nPET added, the turbidity of the water increased by approximately 9.17 NTU. The post–measurement turbidity of different groups was found to return to normal values after a one–day cyclic experiment. Previous research has demonstrated that MPs can stimulate microalgae to produce more EPS [30], which is helpful for the flocculation adsorption of MPs [31]. As such, it can be deduced that MBGS produces EPS to counter the adverse environmental conditions stimulated by the addition of nPET. EPS reduced the concentration of nPET in wastewater through flocculation adsorption,

thus reducing the detrimental effect of shading. However, the production of EPS was limited. When the concentration of nPET was so high that the EPS produced by MBGS were insufficient to absolutely adsorb the nPET in the wastewater, the turbidity of wastewater would rise, which in turn affected the photosynthesis of algae, leading to a reduction in the removal of pollutants. Accordingly, controlling the concentration of nPET is crucial to ensure the normal photosynthesis of algae and the efficient removal of pollutants when treating nPET–containing wastewater using MBGS.

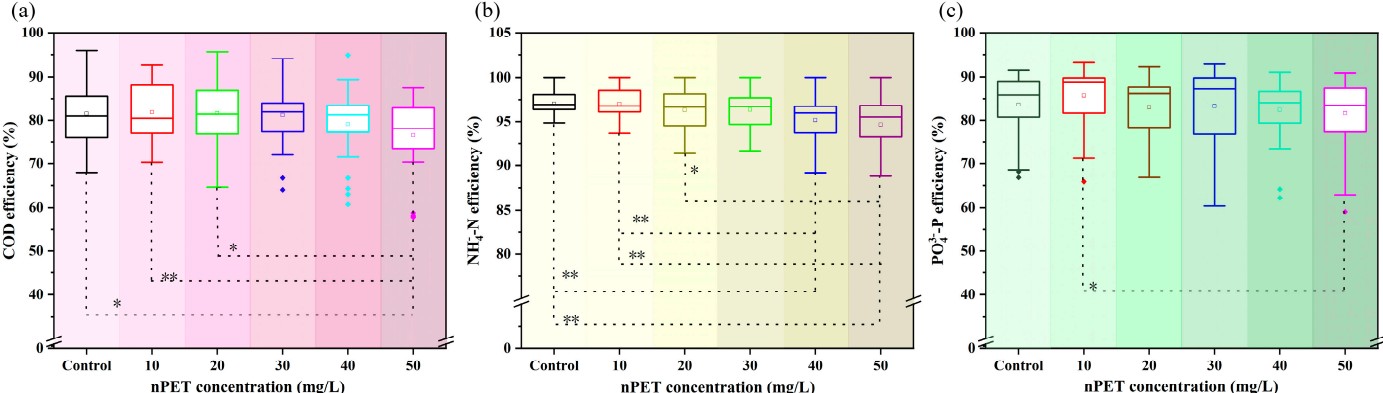

**Figure 1.** Removal efficiencies of COD (**a**), $NH_4^+$–N (**b**), and $PO_4^{3-}$–P (**c**) at different concentrations of nPET, * symbols for $p < 0.05$ and ** symbols for $p < 0.01$.

**Table 1.** Turbidity of the initial intake and final discharge of nPET–containing wastewater.

| Concentration (mg/L) | Control | 10 | 20 | 30 | 40 | 50 |
|---|---|---|---|---|---|---|
| Initial turbidity (NTU) | 8.3 | 27.8 | 35.4 | 44.9 | 53.5 | 64.6 |
| Final turbidity (NTU) | 23.1 | 23.3 | 26.4 | 29 | 29.4 | 28.4 |

From Figure 2, it can be observed that both the pH value and DO concentration of the effluent gradually decreased with an increasing concentration of nPET. Moreover, compared to the control group, both the pH value and DO concentration of the effluent decreased to their lowest values at 50 mg/L nPET, being reduced by 0.60% and 11.01%, respectively. A significant difference in the pH appeared between the control group and the high concentration group of 50 mg/L nPET (Figure 2a, $p < 0.05$), while acetate exhaustion could increase the effluent pH values of MBGS system [32]. This suggests that high concentrations of nPET could inhibit the removal of acetate in this study. As for DO (Figure 2b), there was also a significant difference between the high concentration groups of nPET and the control group ($p < 0.05$), indicating that high concentrations of nPET inhibit the generation of oxygen via photosynthesis. Meanwhile, as can be seen in Table 2, the DO concentration of 50 mg/L nPET–containing wastewater decreased significantly as the experiment progressed, while the DO concentration of wastewater containing low concentrations of nPET ($\leq$30 mg/L) was not affected. Therefore, it can be concluded that low concentrations of nPET appear to have a relatively positive effect on MBGS. Specifically, appropriate stimulation from nPET may enhance the photosynthetic efficiency. Overall, MBGS could have the potential to adapt to the nPET–containing wastewater.

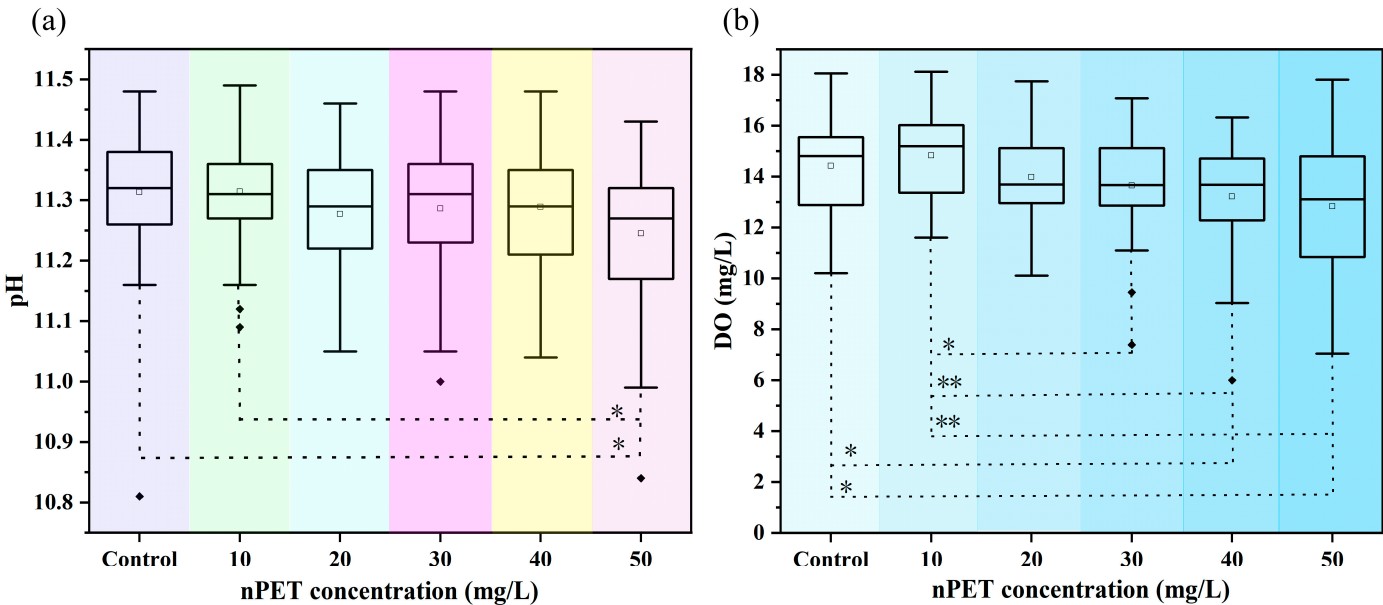

**Figure 2.** The pH (**a**) and the concentration of DO (**b**) at different concentrations of nPET. * Symbols for $p < 0.05$ and ** symbols for $p < 0.01$.

**Table 2.** Dissolved oxygen concentrations of the effluent in different concentration groups of nPET–containing wastewater. Control Group, Group 1, Group 2, Group 3, Group 4, and Group 5 stand for 0, 10, 20, 30, 40, and 50 mg/L nPET, and each group has two parallel samples.

| Time (d) | DO Concentrations of the Effluent (mg/L) | | | | | | | | | | | |
|---|---|---|---|---|---|---|---|---|---|---|---|---|
| | Control Group | | Group 1 | | Group 2 | | Group 3 | | Group 4 | | Group 5 | |
| 1 | 12.43 | 16.75 | 15.75 | 15.50 | 15.12 | 14.70 | 13.55 | 17.08 | 15.48 | 15.40 | 14.80 | 14.65 |
| 2 | 14.80 | 15.30 | 18.03 | 16.61 | 16.49 | 16.20 | 15.62 | 15.38 | 16.32 | 15.41 | 16.13 | 16.20 |
| 3 | 18.06 | 17.15 | 18.12 | 15.70 | 13.05 | 16.90 | 13.15 | 13.68 | 16.15 | 14.30 | 13.03 | 17.65 |
| 4 | 13.75 | 13.35 | 11.75 | 12.65 | 10.11 | 10.70 | 9.45 | 7.39 | 6.00 | 9.03 | 7.05 | 8.20 |
| 5 | 17.14 | 15.55 | 16.05 | 14.13 | 15.50 | 15.00 | 13.65 | 15.07 | 14.51 | 14.48 | 14.33 | 14.00 |
| 6 | 14.32 | 13.36 | 14.40 | 13.37 | 13.56 | 11.11 | 11.10 | 11.12 | 10.64 | 9.75 | 9.69 | 11.16 |
| 7 | 12.37 | 15.28 | 14.34 | 14.00 | 14.42 | 12.13 | 15.12 | 14.64 | 14.36 | 12.98 | 13.20 | 14.35 |
| 8 | 13.20 | 12.61 | 15.00 | 16.02 | 14.90 | 15.14 | 14.10 | 13.25 | 12.65 | 13.46 | 12.66 | 11.00 |
| 9 | 11.30 | 11.28 | 13.59 | 11.90 | 14.43 | 11.92 | 12.86 | 12.88 | 12.29 | 13.55 | 12.38 | 12.81 |
| 10 | 15.48 | 15.20 | 15.55 | 13.25 | 13.20 | 13.06 | 11.32 | 11.92 | 12.08 | 10.10 | 9.38 | 10.40 |
| 11 | 10.21 | 11.85 | 11.60 | 12.54 | 13.07 | 12.10 | 14.52 | 16.25 | 13.65 | 14.83 | 15.77 | 15.08 |
| 12 | 17.15 | 14.96 | 17.60 | 15.89 | 17.75 | 17.45 | 16.26 | 13.98 | 13.72 | 14.58 | 13.20 | 13.30 |
| 13 | 12.88 | 14.54 | 15.56 | 15.40 | 13.47 | 13.45 | 16.61 | 16.66 | 14.71 | 14.69 | 16.13 | 17.81 |
| 14 | 15.00 | 15.70 | 17.41 | 16.41 | 14.87 | 12.96 | 12.86 | 14.32 | 14.91 | 12.28 | 11.44 | 10.84 |
| 15 | 17.03 | 14.82 | 14.06 | 12.58 | 13.83 | 12.90 | 12.25 | 13.49 | 12.57 | 11.55 | 10.65 | 7.53 |

### 3.2. Granule Size and Morphology

Figure 3a indicates that there was a general trend for a decrease in the granule size with an increasing concentration of nPET within the concentration range of 0–50 mg/L nPET. Compared to the control group, it can be observed that the granule size decreased to 93.56%, 86.09%, 93.23%, 84.02%, and 88.61% of the initial one at nPET concentrations of 10, 20, 30, 40, and 50 mg/L, respectively. It was found that MPs could inhibit the growth of algae, and this phenomenon became more and more apparent as the number of MPs increased [33]. A plausible reason for this phenomenon is that the surface of MBGS became looser (as can be observed in SEM) with an increasing concentration of nPET, which made it unfavorable for MBGS to keep the granule state.

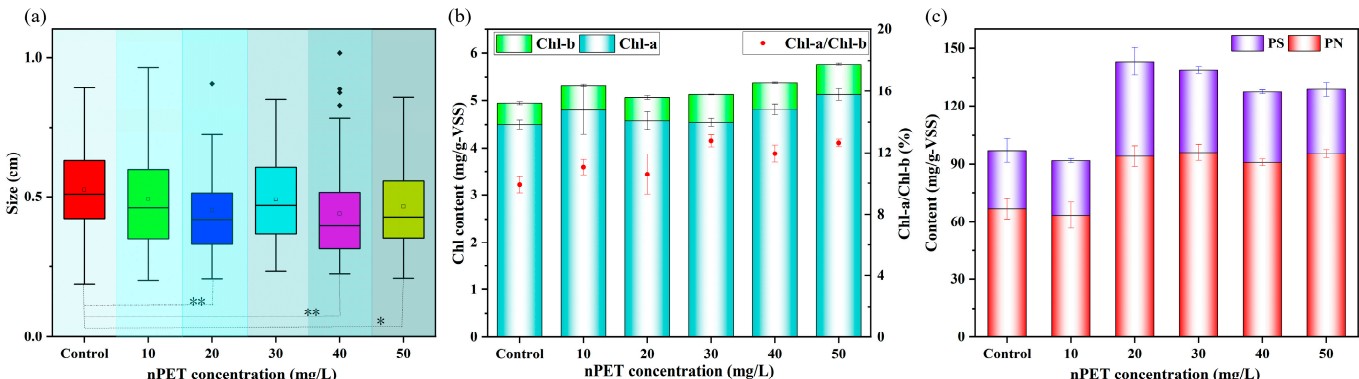

**Figure 3.** Granule size (**a**), contents of Chl (**b**), and EPS (**c**) at different concentrations of nPET.
* Symbols for $p < 0.05$ and ** symbols for $p < 0.01$.

Figure 3b suggests that MBGS might stimulate the production of more Chl to maintain overall photosynthesis and better adapt to highly turbid water. The experiment revealed that the concentration group of 50 mg/L nPET showed increases of 12% in both chlorophyll–a (Chl–a) and chlorophyll–b (Chl–b) compared to the control group. Additionally, the ratio of Chl–a/Chl–b increased. In the Bay of Bengal in India, it was found that the higher the fiber content in the water, the higher the Chl content, but this did not mean that the high fiber area of the algae photosynthetic activity increased [34]. In summary, although the content of Chl increased in wastewater with high concentrations of nPET, and the general photosynthetic efficiency of MBGS may still be reduced.

PN and PS are conducive to the growth of MBGS as they are important components of EPS [35]. Figure 3c shows that the concentrations of PN increased by −4.90%, 41.26%, 44.21%, 36.31%, and 43.30% in 10, 20, 30, 40, and 50 mg/L nPET, respectively, compared to the control group's concentration. This means that PN significantly increased when the concentration of nPET exceeded 10 mg/L, indicating that MBGS secreted more PN to resist the toxic effects of nPET. Excessive production of EPS can effectively resist the influence of toxic substances [36,37], which may promote the adaptability of MBGS in treating nPET–containing wastewater.

Figure 4 shows that the surface of MBGS was covered with numerous nPETs after 15 days, as the number of small white dots around MBGS increased with an increasing concentration of nPET–containing wastewater. The SEM images indicate that the surface of MBGS seemed to become more wrinkled with an increasing concentration of nPET. It was also reported that MPs could wrap around the surface of microalgae according to the SEM images [38]. Therefore, it can be speculated that high concentrations of nPET increase the production of EPS, which in turn allows more nPET encapsulated on the surface of MBGS to be adsorbed, further affecting the physiological activity of MBGS.

*3.3. EPS Spectral Analysis*

From Figure 5a, it can be observed that the different curves represent the FTIR spectra of EPS powder treated with different concentrations of nPET–containing wastewater after 15 days. According to previous studies on the peak frequency of the Fourier transform infrared spectra of biological tissues, the functional groups corresponding to the different peaks can be found [39]. There were significant increases in the halide functional groups at around 482 and 623 $cm^{-1}$, which could be attributed to the presence of halides in the synthetic wastewater used in this experiment. The peak at 1080 $cm^{-1}$ was the main peak of C–O, while the main peaks of C=O were located at 1618 and 1638 $cm^{-1}$. The range of 3200 to 3600 $cm^{-1}$ represents various O–H functional groups. It is evident from the graph that all of these peaks weaken with an increasing concentration of nPET. Due to this, a reasonable hypothesis is that the nPET stimulated MBGS to secrete more EPS, thus providing more surface functional groups, such as hydroxyl, carboxyl, and amino groups,

as adsorption sites. This undoubtedly improved the ability of MBGS to polymerize nPET via adsorption [40]. Due to the occupation of functional group adsorption sites by nPET, more EPS was produced as the concentration of nPET–containing wastewater became higher. Consequently, MBGS could probably remove nPET from wastewater through adsorption.

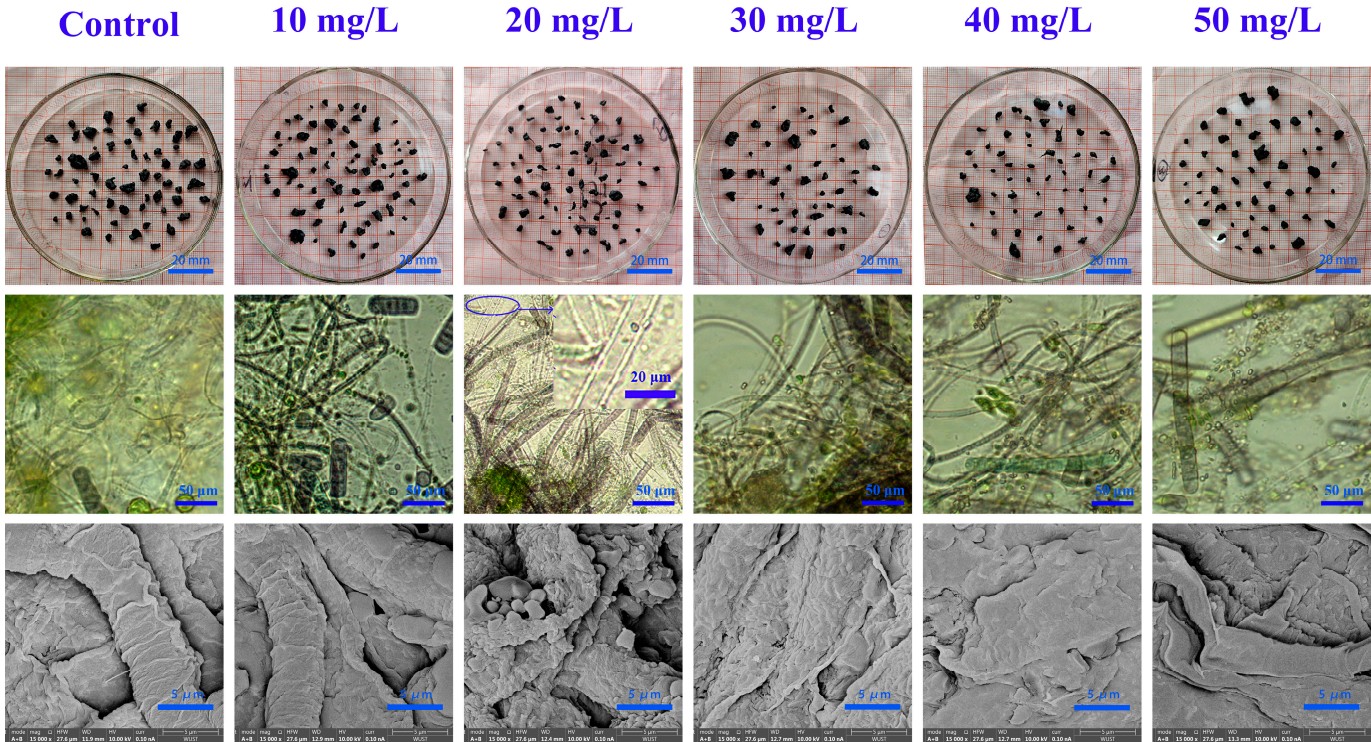

**Figure 4.** Granule morphology, structure, and microbial composition of MBGS with different concentrations of nPET, as determined by digital camera, optical microscope, and SEM.

The XRD spectrum of EPS bound to nPET is shown in Figure 5b. The 2θ values at 31.8°, 45.6°, 56.6°, 66.3°, and 75.3° were consistent with the amorphous nature of EPS aggregates [41]. The widths of the peaks at 31.8°, 45.6°, 56.6°, 66.3°, and 75.3° significantly increased, and the peak intensity decreased with an increasing concentration of nPET, indicating a decrease in the crystallinity of EPS as the concentration of nPET increased. This may be attributed to the mass attachment increase in nPET on the surface of EPS. The concentration of nPET significantly altered the XRD peak characteristics of EPS, making the EPS structure looser, especially at the 31.8° 2θ angle. This result also indirectly reflected the formation of biodeposition of nPET in MBGS through the electrostatic and complexation interactions with EPS.

In Figure 5c, the XPS spectrum of EPS is shown. The signals belonging to C1s, N1s, and O1s appear at 284.9, 398.6 and 532.8 eV, respectively [42]. The contents of C and O on EPS increased after treating EPS with different concentrations of nPET for 15 days, indicating the presence of C and O species in the nPET–EPS aggregates. As shown in Figure 5d, the availability of nitrogen sources played a decisive role in the formation of EPS. And, the ratio of C/N might influence the carbon flux to EPS, which further affects the production and composition of EPS in extreme conditions, like exposure to MPs [43]. The ratio of C/N reached a peak when the concentration of nPET reached 10 mg/L. This suggests that 10 mg/L nPET promotes the growth of EPS and the associated microbial community by serving as a carbon source and potentially aiding in bacterial growth [44]. Particularly, when the concentrations of nPET were higher than 10 mg/L, the ratio of C/N decreased as the concentration of nPET increased. This colud be due to the fact that the proportion of nPET in the nPET–EPS aggregation increased with an increasing concentration of nPET, while nPET does not contain nitrogen. In summary, 10 mg/L nPET helped to enhance the

activity and viability of EPS, while high concentrations of nPET can be more detrimental to microbial activity, resulting in impaired cell function and growth inhibition of some functional bacteria.

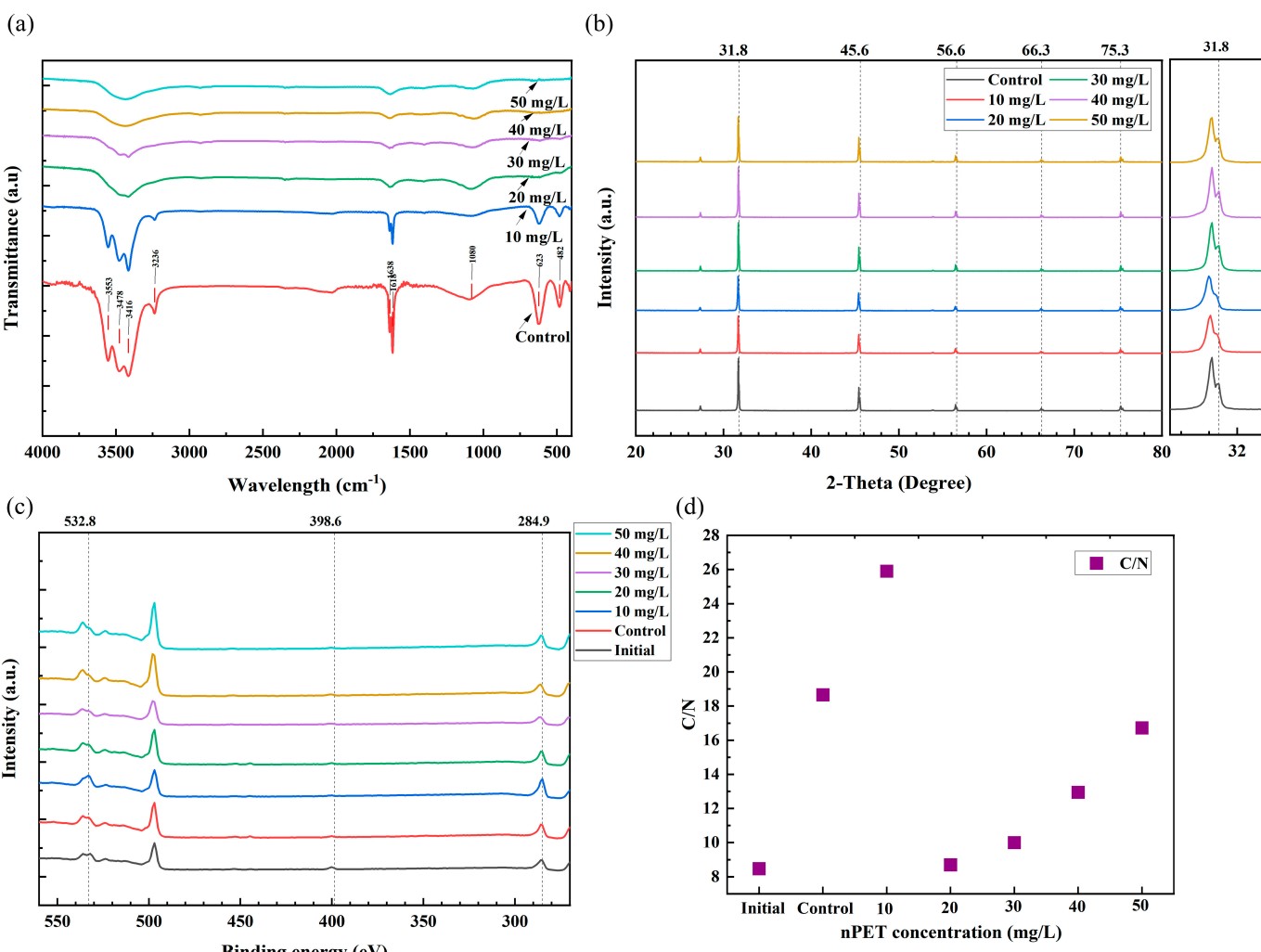

**Figure 5.** Peaks of FTIR (**a**), XRD with a local magnification of 31.8° (**b**), XPS peak diagram (**c**), and the atomic content ratio of C/N (**d**) at different concentrations of nPET.

### 3.4. Microbial Community Analysis

Figure 6 indicates that the concentration of nPET could affect the relative abundance of microorganisms in MBGS. For prokaryotes (Figure 6a,b), the main phyla affected by 50 mg/L nPET were *Proteobacteria*, *Cyanobacteria*, *Chloroflexi*, *Bacteroidetes*, and *Firmicutes*. Specifically, the abundance of *Cyanobacteria* decreased by 23.15% at 50 mg/L nPET compared with the control group. At the family level, *Phormidiaceae* decreased by 32.08%. *Cyanobacteria* play important roles in various physiological activities, especially photosynthesis [45]. The decreased abundance of *Cyanobacteria* could be partly responsible for the decreased DO of 50 mg/L nPET, as shown in Figure 2b. This could be one of the reasons why MBGS was less effective for treating wastewater with high concentrations of nPET. Regarding eukaryotes (Figure 6c,d), it can be seen from Figure 6c that *Chlorophyta* was the dominant population, but there were appreciable differences at the family level. Therefore, to better understand the metabolic pathways of microbial communities, we conducted a functional prediction analysis for microbial communities.

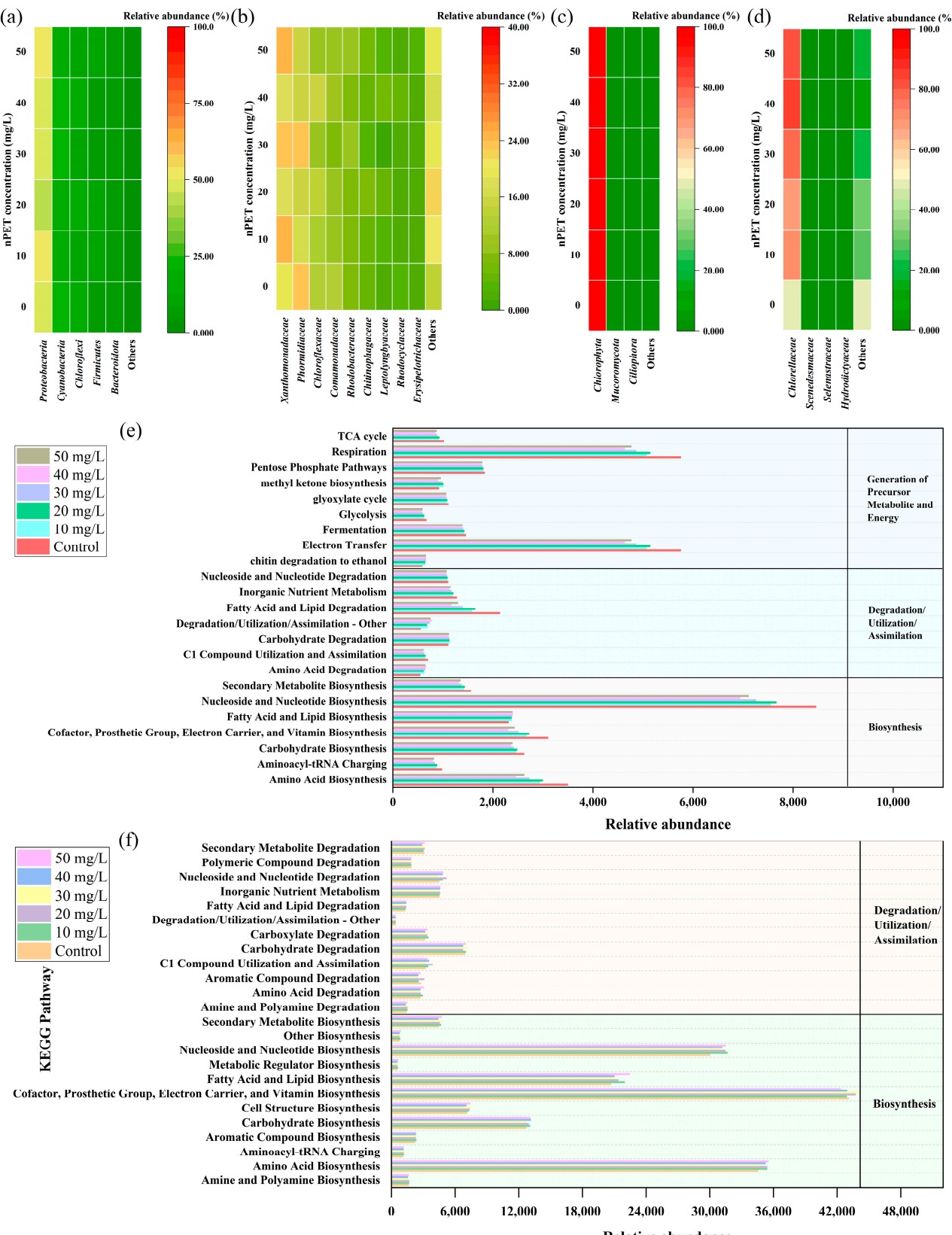

**Figure 6.** The relative abundance of microbes in MBGS at the prokaryote phylum/family level (**a**,**b**) and the eukaryote phylum/family level (**c**,**d**) at different concentrations of nPET; KEGG functional predicted analysis about eukaryotes (**e**) and prokaryotes (**f**).

The functional prediction analysis revealed significant impacts of high concentrations of nPET on the functional profiles of eukaryotic organisms, while the impact on prokaryotic organisms was not significant (Figure 6e,f). From Figure 6e, it can be observed that nPET reduced energy storage materials such as carbohydrates, which suggests that nPET may affect the performance of MBGS by reducing the nutrient intake and storage [46]. At the same time, nPET attached to the surface of MBGS affects electron transfer, which might lead to a reduction in the granule size of MBGS and the removal of COD and $NH_4^+$–N [47]. The decrease in granule size and the pollutant removal of MBGS at high concentrations of nPET might be attributed to inadequate uptake of cellular nutrients caused by physiological and metabolic toxicity.

### 4. Conclusions

As a green process, MBGS exhibited adaptation to nPET–containing wastewater. When the concentration of nPET was less than 30 mg/L, MBGS was almost not affected in the removal of pollutants. Nevertheless, at 50 mg/L nPET, the removal efficiencies of COD and $PO_4^{3-}$–P were slightly affected. In addition, nPET could affect the photosynthesis of microalgae and alter the structure and function of microorganisms. However, MBGS might produce more EPS to protect microbial cells from nPET damage. Overall, MBGS produced an excellent performance and showed good adaptability to nPET–containing wastewater.

**Author Contributions:** C.D.: conceptualization, investigation, writing—original draft, data curation and software. W.X.: investigation, writing—original draft, and data curation. G.Z.: writing—original draft and writing—review & editing. B.J.: funding acquisition and writing—review and editing. All authors have read and agreed to the published version of the manuscript.

**Funding:** This research was funded by the National Natural Science Foundation of China (51808416).

**Data Availability Statement:** Experimental data can be provided as required.

**Conflicts of Interest:** The authors declare no conflict of interest.

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
