# Peer review of "Impact of Nano–Sized Polyethylene Terephthalate on Microalgal–Bacterial Granular Sludge in Non–Aerated Wastewater Treatment"

_water, doi:10.3390/w15223914_

Round 1
Reviewer 1 Report
Comments and Suggestions for Authors
1- why used turbidity (NTU) as indication of treated wastewater?
2- what are type of suitable treatment process selected to treat this type of polluted waste water
3- XPS spectrum used in this study, why?
The literature assessment section can be enriched with recent and relevant works. Following article may become useful to authors: https://doi.org/10.1016/j.heliyon.2023.e12888
5- English should be improved throughout the manuscript.
6- Add a comparative study table based on the literature review
Comments on the Quality of English LanguageMinor editing of English language required
Author Response
Review Comments One
We would like to thank you for reviewing our manuscript. Your comments for our paper have been of great help to us. And we have made changes to various errors and areas that should be corrected in the paper. We summarize our responses to each comment, please review them. We hope our revised manuscript can be accepted for publication.
- why used turbidity (NTU) as indication of treated wastewater?
In this article, turbity is not used as an indicator of treated wastewater, but as a selection criterion for the gradient of the experiment in this study. As shown in Table 1, the turbidity of nPET-containing wastewater was linearly correlated with the concentration of nPET in the wastewater. Every 10 mg/L of nPET addition increased the turbidity of the water by approximately 9.17 NTU, and the turbidity can significantly affect the photosynthesis of microalgae, and the turbidity can significantly affect the photosynthesis of microalgae. Therefore, we used turbidity to select the appropriate n-PET concentration gradient. Furthermore, in this experiment, we used COD, NH4+-N and PO43+-P as indication of treated wastewater.
- what are type of suitable treatment process selected to treat this type of polluted waste water?
We believe that microalgal-bacterial granular sludge (MBGS) is a suitable treatment process to treat municipal wastewater. We conducted this study because municipal wastewater contains microplastics. In the article, we were also able to show that our process can adapt to most concentrations of PET, which means that the current PET concentration in daily life does not affect the normal use of MBGS. At the same time, MBGS process attracted great attention for its high efficiency, low energy consumption, little carbon dioxide emission and huge resource recovery potential.
- XPS spectrum used in this study, why?
The purpose of using XPS analysis is to further analyze the C/N of nPET-EPS through XPS, so as to verify whether nPET promotes or inhibits the activity of MBGS by regulating the synthesis of EPS.
- The literature assessment section can be enriched with recent and relevant works. Following article may become useful to authors: https://doi.org/10.1016/j.heliyon.2023.e12888
Thank you very much for your supplementary recommendation to my literature. I have read this literature carefully. Based on your suggestions I have quoted this article in the appropriate places in my article.
- English should be improved throughout the manuscript.
Thanks for your kind reminding, All the authors have carefully revised this article carefully throughout the manuscript. Thank you.
- Add a comparative study table based on the literature review.
Thank you for this valuable suggestion. Since there have been only a few literatures reported on this area, it may not be suitable to be described in a table. However, your suggestion provides a good idea for my future writing, thank you.
Reviewer 2 Report
Comments and Suggestions for Authors
REVIEW: Impact of nano-sized polyethylene terephthalate on microalgal bacterial granular sludge in non-aerated wastewater Minor revisions are recommended.
ID: water-2663131
The paper presents a theme of great applicability within the WWTP.
Insert in the introductory chapter a section on the impact of MP and NP on purification treatments, on thermophilic sludge. For example, add a paragraph on possible variations in the mechanical behavior of the mud. Effect of increased turbidity and solids concentration on a granular sludge treatment plant. ( https://doi.org/10.3390/app12105198, https://doi.org/10.1016/S0273-1223(97)00672-0).
In the materials and methods paragraph, enter the uncertainty value of the measurements as done for pH.
The possible toxicity of wastewater towards increasing concentrations of PET is studied. Due importance is not given to the actual removal and/or degradation of PET in aqueous solution. Also increase this part as future development. (https://doi.org/10.3390/environments10070108).
Comments on the Quality of English LanguageEnglish is good.
Author Response
Review Comments Two
The paper presents a theme of great applicability within the WWTP.
First of all, thank you very much for your review of our paper. We have carefully read your comments, and your suggestions have greatly improved the quality of our paper. Based on your suggestions, we have made modifications to the paper.
Insert in the introductory chapter a section on the impact of MP and NP on purification treatments, on thermophilic sludge. For example, add a paragraph on possible variations in the mechanical behavior of the mud. Effect of increased turbidity and solids concentration on a granular sludge treatment plant. ( https://doi.org/10.3390/app12105198, https://doi.org/10.1016/S0273-1223(97)00672-0).
Thank you very much for your supplementary recommendation to my literature. I have read this literature carefully. I think your advice is very helpful to me, based on your suggestions I have quoted this article in the appropriate places in my article.
In the materials and methods paragraph, enter the uncertainty value of the measurements as done for pH.
Through your suggestions, I have found out about the description in the paper, and through careful review of the paper, I have solved your suggestions.
The possible toxicity of wastewater towards increasing concentrations of PET is studied. Due importance is not given to the actual removal and/or degradation of PET in aqueous solution. Also increase this part as future development. (https://doi.org/10.3390/environments10070108).
I think your opinion is very far-sighted, the paper you recommended is excellent, I have included it in my paper. At the same time, due to the limited laboratory and individual level, it is difficult to monitor the actual removal and/or degradation of PET in aqueous solution, we also hope to increase the research on nPET through this paper, and lay the foundation for further research in the future.
Reviewer 3 Report
Comments and Suggestions for Authors
This research provides information about the Impact of nano-sized polyethylene terephthalate on microalgal-2 bacterial granular sludge in non-aerated wastewater.
There are small corrections that should be made, observations that have been commented on in the attached document.
The References chapter does not comply with the requirements of the journal.
example: for volume must be italic

Author Response
We than the Reviewer for his/her kind and professional suggestions, according to which the manuscript has been carefully revised.